# Processing of Al/SiC/Gr Hybrid Composite on EDM by Different Electrode Materials Using RSM-COPRAS Approach

**Adel T. Abbas** [1,*], **Neeraj Sharma** [2], **Zeyad A. Alsuhaibani** [1], **Vishal S. Sharma** [3], **Mahmoud S. Soliman** [1] and **Rakesh Chandmal Sharma** [4]

1 Mechanical Engineering Department, College of Engineering, King Saud University, P.O. Box 800, Riyadh 11421, Saudi Arabia; zeyads@ksu.edu.sa (Z.A.A.); solimanm@ksu.edu.sa (M.S.S.)
2 Department of Mechanical Engineering, Maharishi Markandeshwar Engineering College, Maharishi Markandeshwar (Deemed to be University), Mullana, Ambala 133207, India; neerajsharma@mmumullana.org
3 Department of Industrial and Production Engineering, Dr. B. R. Ambedkar National Institute of Technology, Jalandhar 144011, Punjab, India; sharmavs@nitj.ac.in
4 Mechanical Engineering Department, Graphic Era (Deemed to be University), Dehradun 248002, Uttarakhand, India; rcsharmaiitr@gmail.com
* Correspondence: aabbas@ksu.edu.sa

**Abstract:** The present research used the stir-casting method to develop an Al-based composite. The developed composite exhibited challenges while being processed on conventional machining. Thus, a non-traditional machining process was opted to process the composite. The machining variables selected for the current research were the pulse off time (Toff), pulse on time (Ton), servo voltage (SV), current (I), and tool electrode. Three tool electrodes (SS-304, copper, and brass) were used to process the developed composite (Al/SiC/Gr). The experimental plan was designed using response surface methodology (RSM). The output responses recorded for the analysis were the material removal rate (MRR) and tool wear rate (TWR). The obtained data was optimized using complex proportional assessment (COPRAS) and machine learning methods. The optimized settings predicted by the RSM–COPRAS method were Ton: 60 μs; Toff: 60 μs; SV: 7 V; I: 12 A; and tool: brass. The maximum MRR and TWR at the suggested settings were 1.11 g/s and 0.0114 g/s, respectively. A morphological investigation of the machined surface and tool surface was conducted with scanning electron microscopy. The morphological examination of the surface (machined) presented the presence of cracks, lumps, etc.

**Keywords:** Al/SiC/Gr composite; COPRAS; EDM; hybrid composite; optimization

## 1. Introduction

A mixture, which constitutes two diverse materials, is termed a composite, and the properties of a composite are an aggregate of its constituent parts. In the current scenario, metal matrix composites (MMCs) are gaining popularity among industrialists and researchers because of the properties they possess, such as superior fatigue resistance, expansibility, better electrical/thermal conductivity, high microplastic strain resistance, and an improved modulus of electricity [1]. Various materials and their alloys are available and can be utilized as a matrix, such as Zn, Co, Pb, Ni, Ti, Mg, Cu, and Al metals. Of these available materials, Al and its alloy are widely used because of their low weight, high heat conductivity, and easy availability. These alloys have numerous applications in aerospace. To use these alloys, it is essential to cut them into the proper shape. However, due to the hard particles reinforced in the matrix material, a non-traditional machining process is adopted for the machining purpose.

Out of all the available non-traditional techniques, electrical discharge machining (EDM) is considered one of the most viable options due to the nature of the workpiece (Al is conductive in nature). EDM is assumed to be a complex machining process due to the

involvement of many input variables, namely water pressure [2], the material of the electrode [3–7], spark variables (pulse duration, frequency, current, servo voltage, and pulse off time (Toff)) [8], the dielectrics used in the EDM process [9], etc. Considering the above facts, it is essential to perform an EDM control factor performance analysis. Thus, to apply EDM technology successfully to machine hard materials and improve processing, it is mandatory to select appropriate process parameters. Furthermore, improvements in cutting efficiency, cost, productivity, and quality can be made by selecting optimal machining conditions. The parameters are optimized according to the response variables. Sometimes, more than one response variable is considered collectively; these variables are opposite in nature. These kinds of problems are termed multi-criteria decision-making (MCDM) problems. There are many statistical and artificial intelligence techniques, such as response surface methodology (RSM) [10–12] and artificial neural networking (ANN) [13], that are used for modeling purposes. Other techniques, such as the Taguchi method [14–17], Grey relational analysis (GRA) [18–20], desirability [21,22], and principal component analysis (PCA) [23], have been used for performance analysis. To assess the process's machinability, several approaches have been used for planning, modeling, analysis, and optimization [24–28].

From the literature, it is evident that a number of hard-to-machine materials have been processed by EDM [29–31]. These materials include titanium and its alloys, stainless steel, and nickel alloys (Nimonic, Monel, Inconel, and Hastelloy). However, few articles have been published on the machining of Al-based hybrid composites. Due to the reinforcement in Al alloys, it is very difficult to process it with the EDM process (the reinforcements are ceramics, which are non-conductive). Therefore, in the current work, an Al-based hybrid composite was initially developed using a stir-casting route. Then, it was processed with EDM using different process parameters and tool materials. Different electrode materials have different material characteristics; therefore, the output parameters responded differently. The objectives of the current work were:

(i)     To design the experiments as per the number of machining variables and their levels.
(ii)    To investigate the effect of the machining variables on the MRR and TWR during the machining of the Al composite with EDM.
(iii)   To determine the influence of different tool materials on the morphology of the machine surface of Al composite.
(iv)    To establish an empirical relationship between the machining variables and responses after performing experiments.
(v)     To optimize the machining variables of the EDM process using RSM–COPRAS.

## 2. Material Development

### 2.1. Aluminum-Based Metal Matrix Composite (AMC)

The fabrication of an AMC with a uniform distribution of particulate is the greatest challenge for material scientists. The applications of AMC (Al alloys with graphite and SiC) are found in cylinder liners and inertial guidance systems, lightweight optical assemblies, drones, aerospace structures, etc.

These AMCs present a high value of strength with a low weight density. An EDS plot of the developed Al/composite is depicted in Figure 1. These composites may be developed with spray co-deposition, squeeze casting, powder metallurgy, and stir casting. Table 1 shows the composition of the Al composite used in the current work. The AMC was developed using stir casting due to economic factors and availability in the present work. The reinforcements (SiC and Gr) were initially preheated to evaporate moisture. After that, they were mixed with the molten aluminum in a graphite crucible. A graphite stirrer was used to mix the particulates in the molten matrix. Another benefit of preheating the reinforcements was the development of an oxidation layer, which eliminated the chances of forming the $Al_4C_3$ brittle phase. This oxidation layer prevents direct contact between the aluminum (molten) and SiC; thus, the formation of a brittle phase does not occur. At the same time, due to a high degree of graphitization, Gr cannot react with the molten aluminum. The volume fraction, reinforcement size, and shape play pivotal roles in the

mechanical characteristics of AMC. These developed composites exhibit two phases: one is Al with SiC and the other is Al with Gr. The bonding of Al and SiC is stronger than the bonding between Al and Gr.

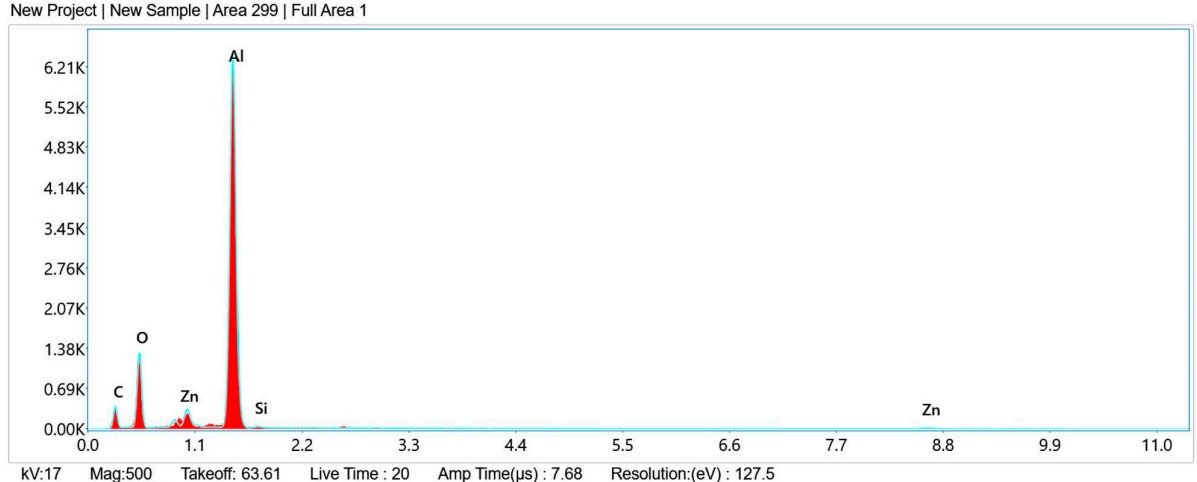

**Figure 1.** EDS plot for developed Al composite.

**Table 1.** Composition of Al composite used in current research.

| Element | Percentage |
| --- | --- |
| Cr | 0.2 |
| Cu | 0.28 |
| Si | 0.6 |
| Mg | 0.74 |
| Zn | 0.14 |
| C | 0.17 |
| Al | Balance |

### 2.2. Experimentations

In the current work, experiments were performed on the developed AMC using the Die-Sinking EDM (Oscar Max, Taichung, Taiwan). The EDM process is shown in Figure 2 with a schematic. Hydrocarbon-based dielectric oil with a density value of 0.76 was used as a dielectric in the current work. There were several process parameters, and out of those, five input parameters at three levels each were selected for the research purpose after a preliminary study. The machining variables used in the current research were pulse on time (Ton), pulse off time (Toff), voltage (V), current (I), and tool material. The machining parameters, along with their symbols, units, and levels, are provided in Table 2.

**Table 2.** Machining parameters and their levels in actual and coded form.

| Machining Parameters | Symbols (Units) | Level/Code | | |
| --- | --- | --- | --- | --- |
| | | −1 | 0 | 1 |
| Tool | | Steel-304 | Brass | Copper |
| Current | I (A) | 10 | 12 | 14 |
| Voltage | V (V) | 6 | 7 | 8 |
| Pulse off time | Toff (μs) | 30 | 60 | 90 |
| Pulse on time | Ton (μs) | 30 | 60 | 90 |

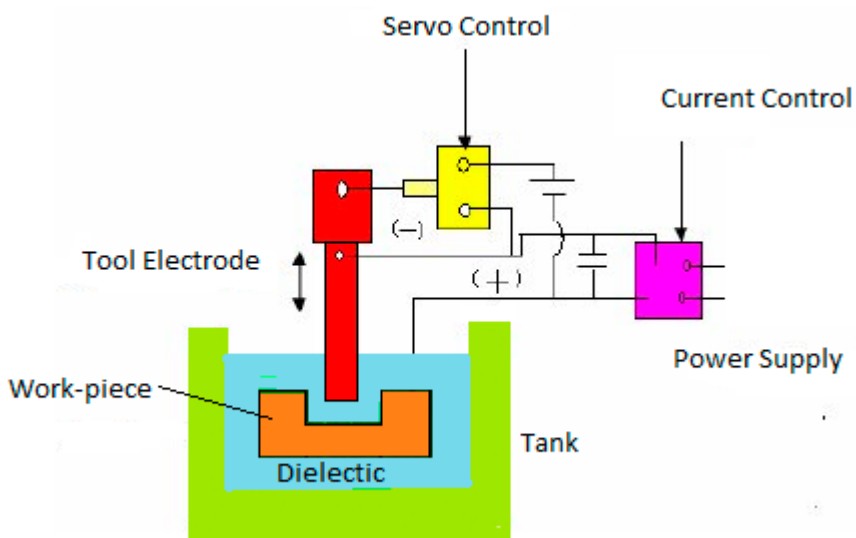

**Figure 2.** Schematic of EDM process.

In the current work, three tool electrodes made of different materials (copper, brass, and steel-304), each with a 12 mm diameter, were used. During the machining (EDM), the workpiece and the tool were submerged in the dielectric. The MRR and TWR were the response variables that were investigated in the current work.

In the EDM, a potential difference was set up between the tool and workpiece; therefore, discrete sparks were generated in the gap. These sparks were sufficient to generate a temperature that melted the material. A hole with a depth of 1 mm was created, and the duration of this was noted. Figure 3 shows the process flow diagram used in the current work.

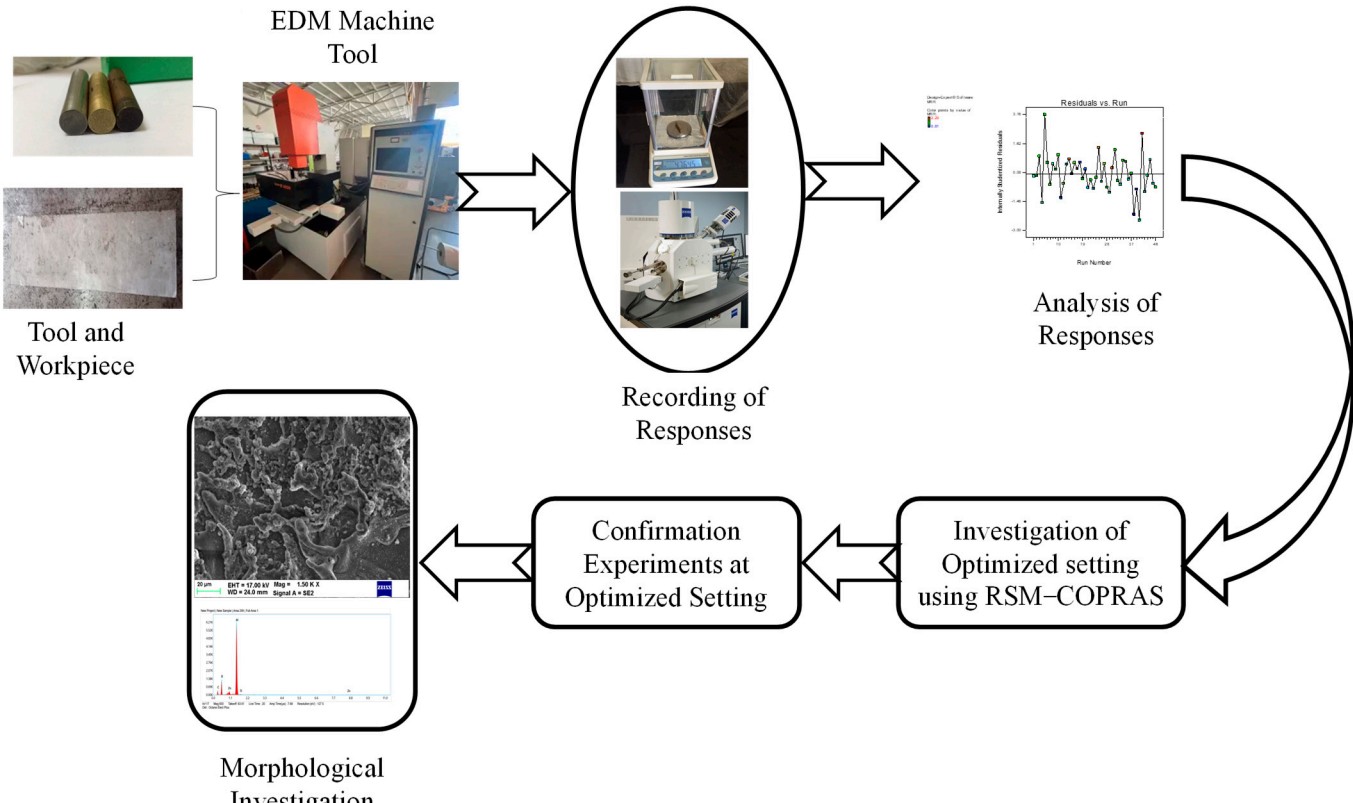

**Figure 3.** Process flow diagram.

### 2.3. Evaluation of Responses

In the current research, the response variables were TWR and MRR. The weight of the tool and workpiece were calculated before and after machining and the time (machining) was also noted down. Equations (1) and (2) were used for the computation of MRR and TWR:

$$\text{MRR}\left(\frac{\text{g}}{\text{s}}\right) = \frac{(W_o - w)}{t_m} \tag{1}$$

where $W_o$—initial workpiece weight (g); W—weight of the workpiece after processing (g); $t_m$—machining time in s for each run;

$$\text{TWR}\left(\frac{\text{g}}{\text{s}}\right) = \frac{(T_o - T)}{t_m} \tag{2}$$

where $T_o$–weight (initial) of the electrode (g); T—weight of the electrode after processing (g). $t_m$—machining time in s for each run.

To compute the weight of the workpieces, a weighing machine with a maximum capacity of 200 g and a lowest count of 0.001 g was used.

## 3. Methodology

The experiments were designed according to RSM-based BBD, and five process parameters with three levels each were used. A total of 46 experiments were designed and executed as per the run order. The experiments were executed as per the run order to check the stability of the machine tool. RSM was used to combine both the statistical and mathematical approaches. It was also beneficial for the investigation of the empirical model and the interaction of the process parameters. There were four steps involved in the implementation of RSM, which were (i) the planning of experiments; (ii) the analysis of the empirical model; (iii) parametric optimization (iv); and the predicted solutions after analysis. RSM also provided the empirical model for the objective function. Equation (3) gives the regression model:

$$Y = f(X1, X2, X3, X4, X5) \tag{3}$$

Here, 'Y' is the response, and 'f' is a function of variables (independent). To study the effect of the control factors on the performance characteristic, an empirical model was developed, which may be linear, two-factor interaction, or quadratic. The general equation of RSM is presented as Equation (4):

$$Y = \beta_o + \sum_{i=1}^{k} \beta_i x_i + \sum_{i=1}^{k} \beta_{ii} x_i^2 + \sum \sum_{i<j} \beta_{ij} x_i x_j + \epsilon \tag{4}$$

The solution obtained after the implementation of RSM was further solved by the COPRAS method, which is described in the paragraph below. The COPRAS method, also known as the complex proportional assessment of alternatives, is a multi-criteria decision-making (MCDM) approach that provides a systematic and structured framework for evaluating and ranking alternatives; it is based on multiple criteria developed by Zavadskas and Turskis in 2002 [32]. COPRAS has gained popularity in various fields, including engineering, management, finance, and environmental sciences.

The COPRAS method aims to address decision-making problems in which multiple criteria need to be considered simultaneously and the alternatives being evaluated have both quantitative and qualitative characteristics. It provides a comprehensive and robust analysis by considering the interdependencies and interactions among the criteria, which is often crucial in complex decision-making scenarios. In this method, the stepwise ranking of experiments was carried out in terms of the utility degree. The benefits of COPRAS are given below [33]:

- In COPRAS, fewer calculations are used compared with TOPSIS and AHP.
- Easy to use compared with TOPSIS.

- It has unique characteristics for maximizing and minimizing the output, which is missing in the maximum technique.
- It can calculate both quantitative and qualitative responses.
- It presents the utility degree, which is the primary benefit compared with other available methods.

The COPRAS method has seven steps, which are described below [34]:

Step 1: In the MCDM-type problem, the first step is to develop the decision matrix as per Equation (5):

$$P = \begin{matrix} A_1 \\ A_2 \\ A_3 \\ . \\ A_m \end{matrix} \begin{bmatrix} P_{11} & P_{12} & P_{13} & . & P_{1n} \\ P_{21} & P_{22} & P_{23} & . & P_{2n} \\ P_{31} & P_{32} & P_{33} & . & P_{3n} \\ . & . & . & . & . \\ P_{m1} & P_{m2} & P_{m3} & . & P_{mn} \end{bmatrix} \tag{5}$$

Step 2: In the second step, normalization was performed, in which all the responses (either in the form of thousands or in the form of decimals) were converted between 0 and 1 using Equation (6):

$$\breve{P}_{ij} = \frac{P_{ij}}{\sum_{i=1}^{m} P_{ij}} \tag{6}$$

Here, $P_{ij}$ is the performance corresponding to the $i$th alternative with respect to the $j$th criterion, $n$ is the number of attributes, and $m$ is the total alternative numbers.

Step 3: In the next step, the weighted normalization was calculated using Equation (7):

$$P' = P_{ij}^* \times W_j \tag{7}$$

Here, the $Wj$ is decided according to the importance of the response variables. If response 1 is given more importance, then its value is kept at 0.7, and the other response is 0.3. Similarly, if response one is given less importance, then its weight should be equal to 0.3, and the other should be 0.7. If both responses are given equal importance, then the weight should be equal to 0.5.

Step 4: Investigation of the minimized and maximized index of all the alternatives.

$$S_{i+} = \sum_{j=1}^{k} P_{ij}(j = 1, 2, 3, \ldots, k) \ maximizing \ index \tag{8}$$

$$S_{i-} = \sum_{j=k+1}^{n} P_{ij}(j = k+1, k+2, \ldots, n) \ minimizing \ index \tag{9}$$

Step 5: Computation of the relative weight of each response using Equation (10):

$$Q_i = S_{+i} + \frac{minS_{-i}\sum_{i=1}^{m} S_{-i}}{S_{-i}\sum_{i=1}^{m} \frac{minS_{-i}}{S_{-i}}} \tag{10}$$

Step 6: Calculation of the ranking of alternatives by comparing their relative weights (Equation (11)). The alternative exhibiting a higher relative weight is given a higher rank (priority)

$$A^* = \{A_i | maxQ_i\} \tag{11}$$

Step 7: Investigation of the performance index for each alternative. *A* value equal to 100 degrees is the best one.

$$P_i = \frac{Q_i}{Q_{max}} \times 100\% \tag{12}$$

## 4. Results and Discussion

The experimental layout was developed by RSM, and the experiments were executed as per the layout provided in Table 3.

**Table 3.** Experimental results using RSM.

| Run | X1: Ton (µs) | X2: Toff (µs) | X3: SV (V) | X4: I (A) | X5: Tool | MRR (g/s) | TWR (g/s) |
|-----|--------------|---------------|------------|-----------|----------|-----------|-----------|
| 1 | 0 | 0 | 0 | 0 | 0 | 1.22 | $5.84 \times 10^{-5}$ |
| 2 | 0 | 0 | 1 | 0 | −1 | 1.58 | 0.000157 |
| 3 | 0 | −1 | 1 | 0 | 0 | 1.59 | 0.00029 |
| 4 | 0 | 0 | 0 | −1 | 1 | 1.24 | $6.19 \times 10^{-5}$ |
| 5 | 0 | 1 | −1 | 0 | 0 | 1.61 | 0.00019 |
| 6 | −1 | 0 | 0 | 0 | −1 | 1.62 | 0.000288 |
| 7 | 0 | 0 | 0 | 1 | −1 | 1.75 | 0.000661 |
| 8 | −1 | 0 | 1 | 0 | 0 | 1.26 | $7.51 \times 10^{-5}$ |
| 9 | 0 | 1 | 0 | 0 | −1 | 1.45 | 0.000106 |
| 10 | 0 | 0 | 0 | −1 | −1 | 1.51 | 0.000121 |
| 11 | −1 | 0 | −1 | 0 | 0 | 0.95 | $2.94 \times 10^{-5}$ |
| 12 | 1 | 0 | 0 | 1 | 0 | 1.79 | 0.000661 |
| 13 | 0 | 0 | 1 | −1 | 0 | 1.08 | $4.30 \times 10^{-5}$ |
| 14 | 0 | 0 | 0 | 1 | 1 | 2.13 | 0.000951 |
| 15 | 0 | 0 | −1 | −1 | 0 | 0.99 | $4.00 \times 10^{-5}$ |
| 16 | −1 | 0 | 0 | 0 | 1 | 1.77 | 0.000661 |
| 17 | 1 | 0 | 0 | 0 | −1 | 2.02 | 0.00074 |
| 18 | 0 | 1 | 0 | −1 | 0 | 0.91 | $2.50 \times 10^{-5}$ |
| 19 | 1 | 0 | 1 | 0 | 0 | 1.58 | 0.000151 |
| 20 | −1 | 0 | 0 | −1 | 0 | 0.96 | $3.40 \times 10^{-5}$ |
| 21 | 0 | 0 | 0 | 0 | 0 | 1.11 | $1.14 \times 10^{-5}$ |
| 22 | 0 | 1 | 0 | 0 | 1 | 1.51 | 0.000117 |
| 23 | 0 | 0 | 0 | 0 | 0 | 1.1 | $4.63 \times 10^{-5}$ |
| 24 | 1 | 0 | −1 | 0 | 0 | 1.59 | 0.000161 |
| 25 | 1 | −1 | 0 | 0 | 0 | 2.04 | 0.000794 |
| 26 | 0 | 1 | 0 | 1 | 0 | 1.24 | $6.45 \times 10^{-5}$ |
| 27 | 0 | 0 | 1 | 0 | 1 | 1.84 | 0.000715 |
| 28 | 0 | 0 | −1 | 0 | −1 | 1.47 | 0.000111 |
| 29 | 0 | 0 | 1 | 1 | 0 | 1.32 | $8.72 \times 10^{-5}$ |
| 30 | 1 | 0 | 0 | 0 | 1 | 2.18 | 0.001039 |
| 31 | 0 | −1 | −1 | 0 | 0 | 1.65 | 0.000313 |
| 32 | 0 | 0 | −1 | 1 | 0 | 1.43 | 0.000105 |
| 33 | −1 | −1 | 0 | 0 | 0 | 1.26 | $8.47 \times 10^{-5}$ |
| 34 | −1 | 0 | 0 | 1 | 0 | 1.54 | 0.000143 |
| 35 | 0 | 0 | 0 | 0 | 0 | 1.36 | 0.000356 |
| 36 | 0 | 0 | 0 | 0 | 0 | 1.19 | $4.76 \times 10^{-5}$ |
| 37 | 0 | −1 | 0 | 1 | 0 | 1.68 | 0.000652 |
| 38 | 0 | −1 | 0 | −1 | 0 | 0.81 | $2.14 \times 10^{-5}$ |
| 39 | −1 | 1 | 0 | 0 | 0 | 0.85 | $2.34 \times 10^{-5}$ |
| 40 | 0 | 0 | −1 | 0 | 1 | 1.32 | $9.38 \times 10^{-5}$ |

**Table 3.** *Cont.*

| Run | X1: Ton (μs) | X2: Toff (μs) | X3: SV (V) | X4: I (A) | X5: Tool | MRR (g/s) | TWR (g/s) |
|---|---|---|---|---|---|---|---|
| 41 | 0 | −1 | 0 | 0 | 1 | 2.29 | 0.001405 |
| 42 | 1 | 1 | 0 | 0 | 0 | 1.28 | $8.53 \times 10^{-5}$ |
| 43 | 1 | 0 | 0 | −1 | 0 | 1.36 | $9.06 \times 10^{-5}$ |
| 44 | 0 | 1 | 1 | 0 | 0 | 1.19 | $6.73 \times 10^{-5}$ |
| 45 | 0 | 0 | 0 | 0 | 0 | 1.15 | $5.61 \times 10^{-5}$ |
| 46 | 0 | −1 | 0 | 0 | −1 | 1.66 | 0.000494 |

*4.1. Analysis of the Responses*

Table 4 presents the ANOVA of MRR and TWR. It is clear from the ANOVA of MRR that the quadratic terms of Ton and tool played an influential role, along with Ton, Toff, and I. However, tool was added forcefully to the model only because the quadratic term tool was significant ($p$-value < 0.05). All terms (except the tool material) presented $p$-values < 0.05. The model of MRR was influential, while the lack of fit was not significant. The R-square, Adj. R-square, Pred. R-square, and adequate prec. were observed in the limits of a good ANOVA. For a good ANOVA, the value of adequate prec. is more than 4; in the present case, it was more than 18, thus presenting a good ANOVA and model. The statistical summary of TWR depicted that the linear terms of Ton, Toff, I, and tool; the interaction terms of Toff and tool; and the polynomial term of tool played pivotal roles in the investigation of TWR. All these terms showed $p$-values of less than 0.05. However, the lack of fit was insignificant, as desired for a better ANOVA. Other terms, namely R2, adj. R2, pred. R2, and adeq. prev., were observed within the limit. From the F-value of process parameters, it was evident that Toff was the major influencing factor, preceded by I, Ton, and tool.

Figure 4 shows the scatter plots and residual plots for MRR and TWR, respectively. Figure 4a shows the scatter plot for MRR, and it is shown in the scatter plot that all the residuals are in a straight line. Thus, it is clear from this plot that the residuals were normally distributed, which is desired for a good model. Another plot for MRR was the residual versus run plot, as represented in Figure 4b. For a good ANOVA, all the residuals must be away from the center line and randomly distributed, which was true in the present case. The scatter plot (Figure 4c) and residual versus run plot (Figure 4d) satisfied all the conditions for a good ANOVA and model. Thus, in the present case, the plots of both responses showed a good ANOVA.

$$\text{MRR} = +1.24108 + 0.22687 * \text{Ton} - 0.18375 * \text{Toff} + 0.25125 * \text{I} + 0.076250 * \text{Tool} + 0.15480 * \text{Ton}^2 + 0.42897 * \text{Tool}^2$$

$$\text{TWR} = +1.60148 \times 10^{-4} + 1.48909 \times 10^{-4} * \text{Ton} - 2.10970 \times 10^{-4} * \text{Toff} + 1.80431 \times 10^{-4} * \text{I} + 1.47856 \times 10^{-4} * \text{Tool} - 2.25200 \times 10^{-4} * \text{Toff} * \text{Tool} + 3.22408 \times 10^{-4} * \text{Tool}^2$$

*4.2. Implementation of COPRAS*

The COPRAS method has seven steps, which are described in the methodology section [34]. In the first step, a decision matrix was developed. Furthermore, the normalization step was completed to convert all the responses between 0 and 1. In the next step, equal importance was given to all the responses, and the weighted normalized value is provided in Table 5. The computation of the relative weight for each response was made, and the results are presented in Table 5. The performance index was calculated, along with the ranking of the process parameter settings.

It is clear from Table 5 that trial run 21 showed the best settings of the process parameters, giving it rank '1' with a Ui value equal to 100. After that, the experimental run numbers 38 and 39 presented Ui values of 55.37 (rank 2) and 51.69 (rank 3), respectively.

Validation experiments were conducted with the proposed optimized settings (Ton: 60; Toff: 60; SV: 7 V and I: 12 A) using all three electrodes (brass, SS-304, and Cu). The

projected values of MRR and TWR are provided in Table 6. With these settings, confirmation experiments indicated that all the experimental values were in close agreement with the predicted solutions. Thus, the proposed approach was successfully applied for the investigation of the optimized settings of EDM while machining the developed Al/SiC/Gr hybrid composite.

**Table 4.** ANOVA for response variables.

| | MRR | | | | | |
|---|---|---|---|---|---|---|
| **Source** | **SS** | **df** | **MS** | **F-Value** | *p*-**Value** | **Remarks** |
| Model | 4.43 | 6 | 0.74 | 21.5 | <0.0001 | significant |
| X1-Ton | 0.82 | 1 | 0.82 | 23.98 | <0.0001 | |
| X2-Toff | 0.54 | 1 | 0.54 | 15.73 | 0.0003 | |
| X4-I | 1.01 | 1 | 1.01 | 29.42 | <0.0001 | |
| X5-Tool | 0.093 | 1 | 0.093 | 2.71 | 0.1078 | |
| $X1^2$ | 0.24 | 1 | 0.24 | 7.12 | 0.0111 | |
| $X5^2$ | 1.88 | 1 | 1.88 | 54.66 | <0.0001 | |
| Residual | 1.34 | 39 | 0.034 | | | |
| Lack of Fit | 1.29 | 34 | 0.038 | 4.14 | 0.0587 | not significant |
| Error | 0.046 | 5 | $9.18 \times 10^{-3}$ | | | |
| Total | 5.77 | 45 | | | | |
| $R^2$ | | 0.767846 | | Pred $R^2$ | 0.679282 | |
| Adj $R^2$ | | 0.73213 | | Adeq Precision | 18.28723 | |
| | TWR | | | | | |
| **Source** | **SS** | **df** | **MS** | **F-Value** | *p*-**Value** | |
| Model | $3.23 \times 10^{-6}$ | 6 | $5.38 \times 10^{-7}$ | 12.7 | <0.0001 | significant |
| X1-Ton | $3.55 \times 10^{-7}$ | 1 | $3.55 \times 10^{-7}$ | 8.38 | 0.0062 | |
| X2-Toff | $7.12 \times 10^{-7}$ | 1 | $7.12 \times 10^{-7}$ | 16.83 | 0.0002 | |
| X4-I | $5.21 \times 10^{-7}$ | 1 | $5.21 \times 10^{-7}$ | 12.31 | 0.0012 | |
| X5-Tool | $3.50 \times 10^{-7}$ | 1 | $3.50 \times 10^{-7}$ | 8.27 | 0.0065 | |
| X2*X5 | $2.03 \times 10^{-7}$ | 1 | $2.03 \times 10^{-7}$ | 4.79 | 0.0346 | |
| $X5^2$ | $1.09 \times 10^{-6}$ | 1 | $1.09 \times 10^{-6}$ | 25.63 | <0.0001 | |
| Residual | $1.65 \times 10^{-6}$ | 39 | $4.23 \times 10^{-7}$ | | | |
| Lack of Fit | $1.57 \times 10^{-6}$ | 34 | $4.61 \times 10^{-7}$ | 2.79 | 0.1266 | not significant |
| Error | $8.26 \times 10^{-7}$ | 5 | $1.65 \times 10^{-7}$ | | | |
| Total | $4.88 \times 10^{-6}$ | 45 | | | | |
| $R^2$ | | 0.661495 | | Pred $R^2$ | 0.464859 | |
| Adj $R^2$ | | 0.609417 | | Adeq Precision | 16.17298 | |

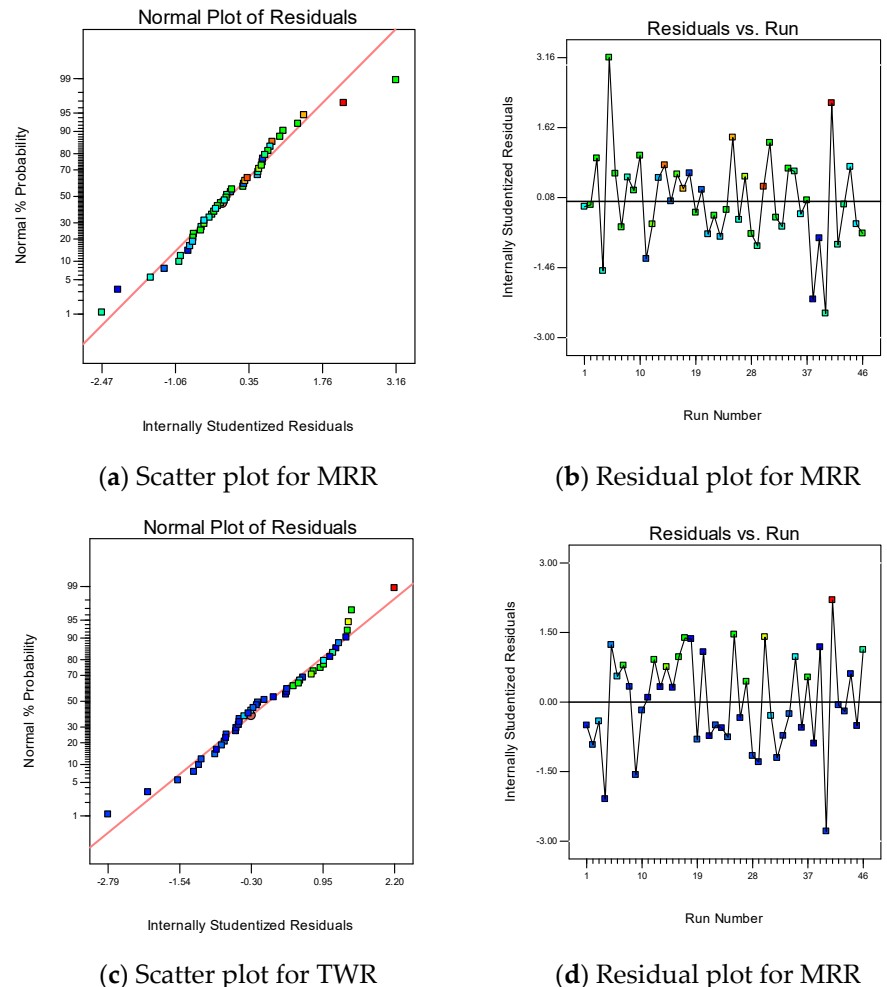

(**a**) Scatter plot for MRR

(**b**) Residual plot for MRR

(**c**) Scatter plot for TWR

(**d**) Residual plot for MRR

**Figure 4.** Plots for MRR and TWR.

**Table 5.** COPRAS implementation.

| Sr No | Data Set | | Normalized Data | | Weighted Normalized Decision Matrix | | Bi | Ci | Min (Ci)/Ci | Qi | Ui | Rank |
|---|---|---|---|---|---|---|---|---|---|---|---|---|
| | MRR | TWR | MRR | TWR | | | | | | | | |
| 1 | 1.22 | $5.84 \times 10^{-5}$ | 0.0184 | 0.0047 | 0.00918 | 0.00233 | 0.009 | 0.002 | 0.1952 | 0.023 | 29.1636 | 12 |
| 2 | 1.58 | 0.00016 | 0.0238 | 0.0125 | 0.01189 | 0.00627 | 0.012 | 0.006 | 0.0726 | 0.017 | 21.6721 | 31 |
| 3 | 1.59 | 0.00029 | 0.0239 | 0.0231 | 0.01197 | 0.01157 | 0.012 | 0.012 | 0.0393 | 0.015 | 18.7933 | 40 |
| 4 | 1.24 | $6.19 \times 10^{-5}$ | 0.0187 | 0.0049 | 0.00933 | 0.00247 | 0.009 | 0.002 | 0.1842 | 0.022 | 28.3698 | 13 |
| 5 | 1.61 | 0.00019 | 0.0242 | 0.0152 | 0.01212 | 0.00758 | 0.012 | 0.008 | 0.0600 | 0.016 | 20.8338 | 35 |
| 6 | 1.62 | 0.00029 | 0.0244 | 0.0230 | 0.01219 | 0.01149 | 0.012 | 0.011 | 0.0396 | 0.015 | 19.1061 | 39 |
| 7 | 1.75 | 0.00066 | 0.0263 | 0.0528 | 0.01317 | 0.02638 | 0.013 | 0.026 | 0.0172 | 0.014 | 18.3601 | 43 |
| 8 | 1.26 | $7.51 \times 10^{-5}$ | 0.0190 | 0.0060 | 0.00948 | 0.00300 | 0.009 | 0.003 | 0.1518 | 0.020 | 25.6703 | 16 |
| 9 | 1.45 | 0.00011 | 0.0218 | 0.0085 | 0.01091 | 0.00423 | 0.011 | 0.004 | 0.1075 | 0.018 | 23.5435 | 22 |
| 10 | 1.51 | 0.00012 | 0.0227 | 0.0097 | 0.01137 | 0.00483 | 0.011 | 0.005 | 0.0942 | 0.018 | 22.9292 | 26 |
| 11 | 0.95 | $2.94 \times 10^{-5}$ | 0.0143 | 0.0023 | 0.00715 | 0.00117 | 0.007 | 0.001 | 0.3878 | 0.034 | 43.7694 | 5 |
| 12 | 1.79 | 0.00066 | 0.0269 | 0.0528 | 0.01347 | 0.02638 | 0.013 | 0.026 | 0.0172 | 0.015 | 18.7446 | 41 |
| 13 | 1.08 | $4.30 \times 10^{-5}$ | 0.0163 | 0.0034 | 0.00813 | 0.00172 | 0.008 | 0.002 | 0.2651 | 0.027 | 34.0633 | 8 |
| 14 | 2.13 | 0.00095 | 0.0321 | 0.0759 | 0.01603 | 0.03795 | 0.016 | 0.038 | 0.0120 | 0.017 | 21.5425 | 33 |
| 15 | 0.99 | $4.00 \times 10^{-5}$ | 0.0149 | 0.0032 | 0.00745 | 0.00160 | 0.007 | 0.002 | 0.2850 | 0.027 | 34.9745 | 7 |

**Table 5.** *Cont.*

| Sr No | Data Set | | Normalized Data | | Weighted Normalized Decision Matrix | | Bi | Ci | Min (Ci)/Ci | Qi | Ui | Rank |
|---|---|---|---|---|---|---|---|---|---|---|---|---|
| | MRR | TWR | MRR | TWR | | | | | | | | |
| 16 | 1.77 | 0.00066 | 0.0266 | 0.0528 | 0.01332 | 0.02638 | 0.013 | 0.026 | 0.0172 | 0.015 | 18.5523 | 42 |
| 17 | 2.02 | 0.00074 | 0.0304 | 0.0591 | 0.01520 | 0.02953 | 0.015 | 0.030 | 0.0154 | 0.016 | 20.7907 | 36 |
| 18 | 0.91 | $2.50 \times 10^{-5}$ | 0.0137 | 0.0020 | 0.00685 | 0.00100 | 0.007 | 0.001 | 0.4560 | 0.039 | 49.4814 | 4 |
| 19 | 1.58 | 0.00015 | 0.0238 | 0.0121 | 0.01189 | 0.00603 | 0.012 | 0.006 | 0.0755 | 0.017 | 21.9298 | 29 |
| 20 | 0.96 | $3.40 \times 10^{-5}$ | 0.0145 | 0.0027 | 0.00723 | 0.00136 | 0.007 | 0.001 | 0.3353 | 0.031 | 39.1791 | 6 |
| 21 | 1.11 | $1.14 \times 10^{-5}$ | 0.0167 | 0.0009 | 0.00835 | 0.00045 | 0.008 | 0.000 | 1.0000 | 0.078 | 100.0000 | 1 |
| 22 | 1.51 | 0.00012 | 0.0227 | 0.0093 | 0.01137 | 0.00467 | 0.011 | 0.005 | 0.0974 | 0.018 | 23.2169 | 25 |
| 23 | 1.1 | $4.63 \times 10^{-5}$ | 0.0166 | 0.0037 | 0.00828 | 0.00185 | 0.008 | 0.002 | 0.2462 | 0.026 | 32.5675 | 10 |
| 24 | 1.59 | 0.00016 | 0.0239 | 0.0129 | 0.01197 | 0.00643 | 0.012 | 0.006 | 0.0708 | 0.017 | 21.6070 | 32 |
| 25 | 2.04 | 0.00079 | 0.0307 | 0.0634 | 0.01535 | 0.03169 | 0.015 | 0.032 | 0.0144 | 0.016 | 20.8893 | 34 |
| 26 | 1.24 | $6.45 \times 10^{-5}$ | 0.0187 | 0.0051 | 0.00933 | 0.00257 | 0.009 | 0.003 | 0.1767 | 0.022 | 27.7066 | 14 |
| 27 | 1.84 | 0.00072 | 0.0277 | 0.0571 | 0.01385 | 0.02854 | 0.014 | 0.029 | 0.0159 | 0.015 | 19.1088 | 38 |
| 28 | 1.47 | 0.00011 | 0.0221 | 0.0089 | 0.01106 | 0.00443 | 0.011 | 0.004 | 0.1027 | 0.018 | 23.3029 | 24 |
| 29 | 1.32 | $8.72 \times 10^{-5}$ | 0.0199 | 0.0070 | 0.00994 | 0.00348 | 0.010 | 0.003 | 0.1307 | 0.019 | 24.3654 | 17 |
| 30 | 2.18 | 0.00104 | 0.0328 | 0.0829 | 0.01641 | 0.04147 | 0.016 | 0.041 | 0.0110 | 0.017 | 21.9324 | 28 |
| 31 | 1.65 | 0.00031 | 0.0248 | 0.0250 | 0.01242 | 0.01249 | 0.012 | 0.012 | 0.0364 | 0.015 | 19.1120 | 37 |
| 32 | 1.43 | 0.00011 | 0.0215 | 0.0084 | 0.01076 | 0.00419 | 0.011 | 0.004 | 0.1086 | 0.018 | 23.4428 | 23 |
| 33 | 1.26 | $8.47 \times 10^{-5}$ | 0.0190 | 0.0068 | 0.00948 | 0.00338 | 0.009 | 0.003 | 0.1346 | 0.019 | 24.1334 | 20 |
| 34 | 1.54 | 0.00014 | 0.0232 | 0.0114 | 0.01159 | 0.00571 | 0.012 | 0.006 | 0.0797 | 0.017 | 21.9227 | 30 |
| 35 | 1.36 | 0.00036 | 0.0205 | 0.0284 | 0.01024 | 0.01421 | 0.010 | 0.014 | 0.0320 | 0.012 | 15.9317 | 46 |
| 36 | 1.19 | $4.76 \times 10^{-5}$ | 0.0179 | 0.0038 | 0.00896 | 0.00190 | 0.009 | 0.002 | 0.2395 | 0.026 | 32.8318 | 9 |
| 37 | 1.68 | 0.00065 | 0.0253 | 0.0520 | 0.01264 | 0.02602 | 0.013 | 0.026 | 0.0175 | 0.014 | 17.7086 | 45 |
| 38 | 0.81 | $2.14 \times 10^{-5}$ | 0.0122 | 0.0017 | 0.00610 | 0.00085 | 0.006 | 0.001 | 0.5327 | 0.043 | 55.3729 | 2 |
| 39 | 0.85 | $2.34 \times 10^{-5}$ | 0.0128 | 0.0019 | 0.00640 | 0.00093 | 0.006 | 0.001 | 0.4872 | 0.040 | 51.6900 | 3 |
| 40 | 1.32 | $9.38 \times 10^{-5}$ | 0.0199 | 0.0075 | 0.00994 | 0.00374 | 0.010 | 0.004 | 0.1215 | 0.018 | 23.5436 | 21 |
| 41 | 2.29 | 0.00141 | 0.0345 | 0.1121 | 0.01724 | 0.05607 | 0.017 | 0.056 | 0.0081 | 0.018 | 22.7343 | 27 |
| 42 | 1.28 | $8.53 \times 10^{-5}$ | 0.0193 | 0.0068 | 0.00963 | 0.00340 | 0.010 | 0.003 | 0.1336 | 0.019 | 24.2410 | 19 |
| 43 | 1.36 | $9.06 \times 10^{-5}$ | 0.0205 | 0.0072 | 0.01024 | 0.00362 | 0.010 | 0.004 | 0.1258 | 0.019 | 24.3115 | 18 |
| 44 | 1.19 | $6.73 \times 10^{-5}$ | 0.0179 | 0.0054 | 0.00896 | 0.00269 | 0.009 | 0.003 | 0.1694 | 0.021 | 26.5692 | 15 |
| 45 | 1.15 | $5.61 \times 10^{-5}$ | 0.0173 | 0.0045 | 0.00866 | 0.00224 | 0.009 | 0.002 | 0.2032 | 0.023 | 29.2057 | 11 |
| 46 | 1.66 | 0.00049 | 0.0250 | 0.0394 | 0.01249 | 0.01972 | 0.012 | 0.020 | 0.0231 | 0.014 | 18.0160 | 44 |

**Table 6.** Confirmation table for MRR and TWR.

| Ton | Toff | V | I | Tool | Predicted | | Experimental | |
|---|---|---|---|---|---|---|---|---|
| | | | | | MRR | TWR | MRR | TWR |
| 60 | 60 | 7 | 12 | Brass | 1.11 | $1.14 \times 10^{-5}$ | 1.03 | 0.00103 |
| 60 | 60 | 7 | 12 | SS-304 | 1.36 | 0.000091 | 1.52 | 0.000098 |
| 60 | 60 | 7 | 12 | Cu | 1.66 | 0.000125 | 1.79 | 0.000137 |

## 5. Morphological Investigations

Discharge energy is generated between the workpiece and the tool, which removes the material from the work surface. The machining parameter settings play an important role in the MRR. A high value of Ton and low values of Toff and SV develop a high

discharge energy, which removes large craters from the surface and increases the MRR value. Similarly, a low value of Ton and high values of SV and Toff develop a low discharge energy. The discharge energy is the main factor that eradicates the material from the work surface. There is always a gap between the workpiece and the tool. A spark is developed between the tool and the workpiece, and this spark is used to remove the material. The high temperature between the spark gap causes the material to melt, and at the same time, the dielectric comes in contact with it, which develops a recast layer and deposits lumps. The alternate heating and cooling cause the generation of microcracks on the surface. The machined surface at 1500× magnification is shown in Figure 5. It is clear from this that the machined surface had sub-surface and deposited lumps, etc.

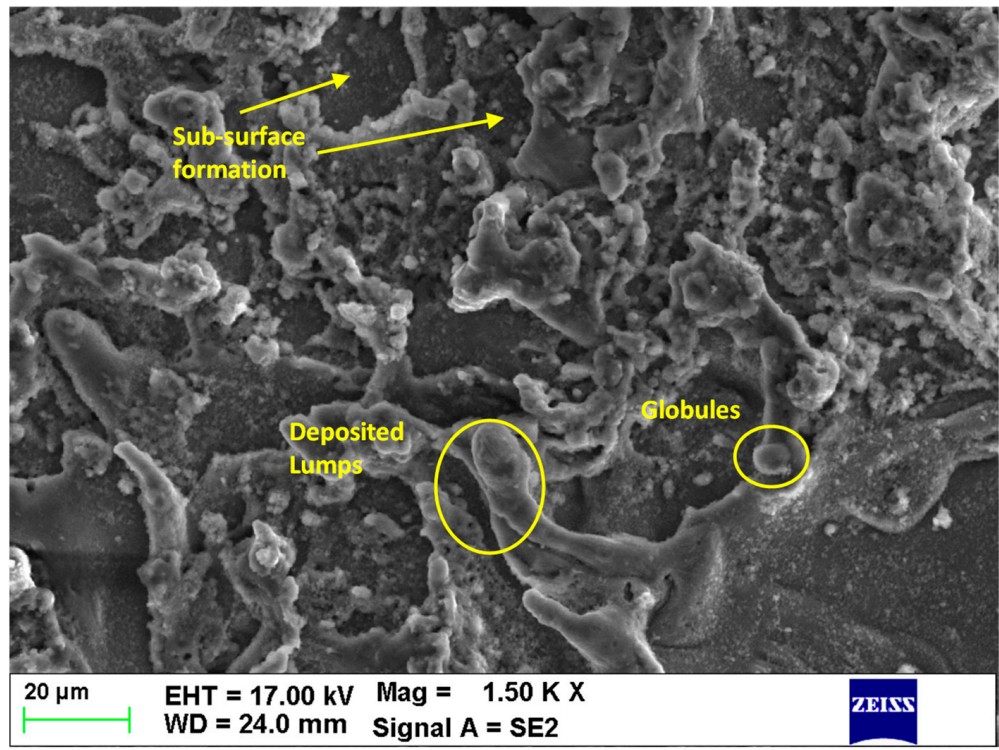

**Figure 5.** SEM images of workpiece after EDM.

Figure 6 represents the SEM micrograph of the SS-304, brass, and copper electrodes after machining the aluminum composite. Figure 6a shows that the SS-304 tool surface had a large number of cracks, along with some crater marks. In the case of the brass electrode (Figure 6b), along with the craters and cracks, some lumps were observed. The presence of lumps indicates the presence of extra material that was not eliminated by the dielectric due to less Toff time. Figure 6c shows a large number of craters and cracks. The tool surface exhibited surface irregularities after machining the composite. The main difference between the three materials is their melting point; SS-304, Cu, and brass have melting points of ~1430 °C, 1085 °C, and 930 °C, respectively. A lower melting point leads to more damage to the tool's surface. Table 6 shows the surface roughness values at the optimized settings provided by the proposed approach, RSM–COPRAS, using different tool materials. It is evident from Table 7 that the surface roughness value for the surface machined by the SS-304 electrode was the highest, which was also clarified by the SEM image (Figure 6a). Similarly, the surfaces machined by brass and Cu exhibited surface roughness values of 2.96 µm and 2.69 µm, respectively.

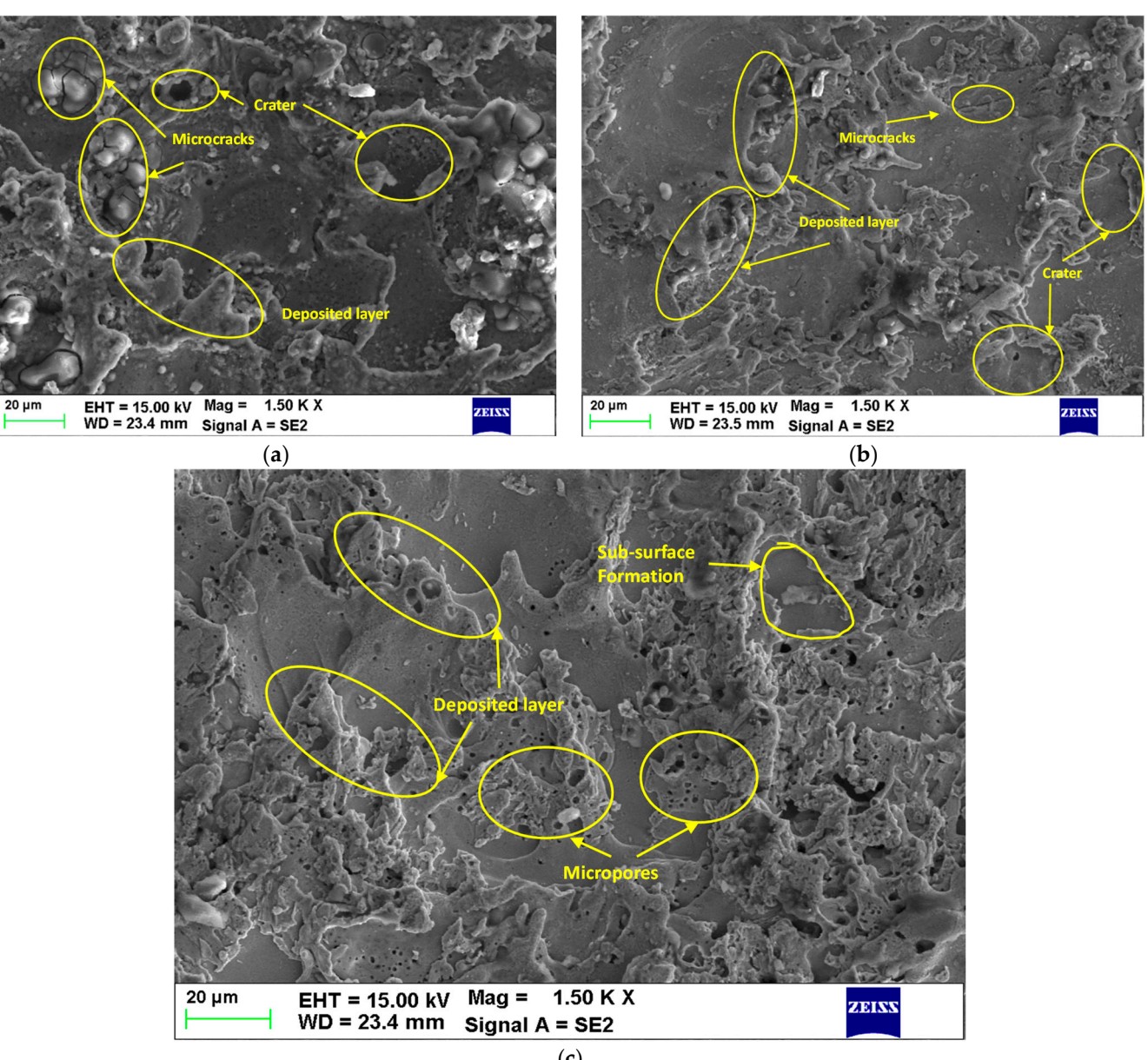

**Figure 6.** SEM micrographs of tool after EDM, (**a**) SS, (**b**) brass, and (**c**) Cu.

**Table 7.** Surface roughness values corresponding to different tool materials.

| Tool Material | Surface Roughness |
| --- | --- |
| SS-304 | 3.19 |
| Brass | 2.96 |
| Copper | 2.67 |

## 6. Concluding Remarks

In the current research, an Al composite was developed by the stir-casting route, and after that, it was processed by EDM using three different types of electrodes. The following conclusions were found by the current research:

1. After the preliminary study, the significant machining parameters of EDM were investigated; these were Ton, Toff, SV, and I.

2. ANOVA showed that I played a pivotal role in MRR, followed by Ton, Toff, and I. Similarly, ANOVA of TWR showed that Toff, I, and Ton had a significant influence.
3. The integrated approach of RSM–COPRAS suggested that the optimized settings for MRR and TWR are Ton: 60; Toff: 60; V: 7; I: 12; and tool: brass. At these settings, the MRR and TWR were 1.11 g/s and 0.0114 g/s, respectively.
4. The morphological investigation revealed the presence of cracks, craters, and lumps on the workpiece and tool. The maximum TWR was observed in the case of brass, followed by those of Cu and SS-304.
5. From this research, it can be found that the current methodology is a very effective and powerful technique to tackle the multi-response problems in industrial experiments.

After careful analysis, it was found that the proposed technique can effectively be applied for the optimization of other conventional and non-conventional processes, such as WEDM, milling, drilling, USM, AJM, AFM, etc. The proposed approach can be used for other responses, such as surface quality, geometrical error, dimensional accuracy, etc.

**Author Contributions:** A.T.A., N.S. and Z.A.A.: investigation, conceptualization, methodology, data curation, validation, visualization; M.S.S.: investigation, writing—original draft, writing; V.S.S.: conceptualization, supervision, project administration; R.C.S.: review and editing; A.T.A.: funding acquisition. All authors have read and agreed to the published version of the manuscript.

**Funding:** Deputyship for Research and Innovation, Ministry of Education, Saudi Arabia Grant (IFKSUOR3–040-1).

**Institutional Review Board Statement:** Not applicable.

**Informed Consent Statement:** Not applicable.

**Data Availability Statement:** Not applicable.

**Acknowledgments:** The authors extend their appreciation to the Deputyship for Research and Innovation, Ministry of Education, Saudi Arabia, for funding this research work through project no. IFKSUOR3–040-1.

**Conflicts of Interest:** The authors declare that they have no known competing financial interests or personal relationships that could have appeared to influence the work reported in this paper.

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
