# Peer review of "Processing of Al/SiC/Gr Hybrid Composite on EDM by Different Electrode Materials Using RSM-COPRAS Approach"

_metals, doi:10.3390/met13061125_

Round 1
Reviewer 1 Report
1) The composition and proportion of the processing target material, as well as the source of the material, should be introduced in the manuscript.
2) In EDM, the composition of oil-based insulating dielectric has a significant impact on the test results, and the information on the dielectric should be introduced in the manuscript.
3)Suggest providing a schematic diagram of EDM processing test methods. The detection methods for MRR and TWR should also be introduced in the manuscript.
4)In Chapter 5,there is a problem with the expression of “The discharge energy generates between the workpiece and tool. There is no direct contact between the workpiece and tool electrode, thus there will be no residual stress.” In EDM, there is a significant residual stress phenomenon on the surface of molten metal after solidification.
5)It is recommended to discuss the surface roughness comparison of the processing results of the three tool materials in the manuscript.
6)In the chapter “Concluding Remarks”,“The stir-casting route successfully develops an Al-based hybrid composite" is not the subject of this manuscript's research.
Author Response
Reviewer-1
Comment 1: The composition and proportion of the processing target material, as well as the source of the material, should be introduced in the manuscript.
Response to comment 1: The composition of the material has been introduced in
Comment 2: In EDM, the composition of oil-based insulating dielectric has a significant impact on the test results, and the information on the dielectric should be introduced in the manuscript.
Response to Comment 2: The hydrocarbon-based dielectric is used in the current work, which is described in section 2.2
Comment 3: Suggest providing a schematic diagram of EDM processing test methods. The detection methods for MRR and TWR should also be introduced in the manuscript.
Response to comment 3: Schematic diagram of EDM has been incorporated
Comment 4: In Chapter 5,there is a problem with the expression of “The discharge energy generates between the workpiece and tool. There is no direct contact between the workpiece and tool electrode, thus there will be no residual stress.” In EDM, there is a significant residual stress phenomenon on the surface of molten metal after solidification.
Response to Comment 4: Thanks for this suggestion, we have modify the sentence.
Comment 5: It is recommended to discuss the surface roughness comparison of the processing results of the three tool materials in the manuscript.
Response to Comment 5: The surface roughness corresponding to three tool material has been discussed in section 5
Comment 6:In the chapter “Concluding Remarks”,“The stir-casting route successfully develops an Al-based hybrid composite" is not the subject of this manuscript's research.
Response to Comment 6: In the conclusion section, the required modification has been made.
Reviewer 2 Report
In general, the paper is characterized by several problems to be accepted for the publication. There is no clarity in the description of the methodologies and the discussion of the results is very poor. In my opinion the paper should be completely revised before the acceptance. In particular, all the sections need to be explained more in details for underlining what the authors what demonstrate, what they do and finally, what they found. The entire manuscript need to be revised from the language point of view, since in some case there typos and in other the sentences are not clear.
Below you can find some more suggestions.
In the introduction several abbreviations are reported but there are no references to their meaning. Please fix this (e.g. GRA, EDM…).
Several works are cited together explaining the concept in just one sentence. Please explain better what the previous research did.
The last part of introduction describes what was done in the manuscript, but the final goal and the novelty are not clear.
In section 2.2 the input parameters are just cited with their signs. Please indicate what they are the first time that appear in the manuscript.
In section 2.2 line 117, the authors cite SV. Which kind of input parameters is it? It is not used for the experimental plan.
Section 3 does not supply all information about the methodologies applied in the research. There are no references to the COPRAS implementation and to the morphological analysis.
Line 174: It should be “statistical” instead of “statistical”.
Section 4.1. In line 170 the authors said that all p-value are lower than 0.05, it is not true. For the MRR the p-value of X5-Tool is higher (0.1078). This means that the Tool does not affect the MRR.
Section 5. The first sentence has no sense (line 252 – “The discharge energy generates between the workpiece and the toll.”). For sake of clarity, I suggest adding to Fig. 5 evidence of cracks, lumps and other details reported in the text.
What can you conclude from the morphological analysis? It is not correlated to different levels of the input, but it is in reported just as a function of the tool. This is because the aspect of the surface is the same varying the other parameters and the only effect that you can see is related to the tool? Did you detect recast particles? Different tool affect the presence of these particles?
PROOF READING IS SUGGESTED
Author Response
Reviewer-2
General comment: There is no clarity in the description of the methodologies and the discussion of the results is very poor.
Response to General Comment: The methodology section has been revised.
Comment 1: In the introduction several abbreviations are reported but there are no references to their meaning. Please fix this (e.g. GRA, EDM…).
Response to comment 1: Full form of all the abbreviation is provided at the first instance and after that abbreviated form is used.
Comment 2: Several works are cited together explaining the concept in just one sentence. Please explain better what the previous research did.
Response to Comment 2: Authors tried to their level best to explain each of the previous work explained separately at most of the places. But still there are few places of combined reference, where methodology is mentioned at some specific area.
Comment 3: The last part of introduction describes what was done in the manuscript, but the final goal and the novelty are not clear.
Response to comment 3: Authors are thankful for this suggestion; the goals and novelty statement has been revised.
Comment 4: In section 2.2 the input parameters are just cited with their signs. Please indicate what they are the first time that appear in the manuscript.
Response to comment 4: The full form is described at first instance and after that full form is used.
Comment 5: In section 2.2 line 117, the authors cite SV. Which kind of input parameters is it? It is not used for the experimental plan.
Response to comment 5: Authors are thankful for this suggestion, the corrections has been made.
Comment 6: Section 3 does not supply all information about the methodologies applied in the research. There are no references to the COPRAS implementation and to the morphological analysis.
Response to comment 6: The COPRAS methodology is incorporated in Section 3. All the steps has been described in section 3 now.
Comment 7: Line 174: It should be “statistical” instead of “statistcal”.
Response to Comment 7: Typos have been fixed.
Comment 8: Section 4.1. In line 170 the authors said that all p-value are lower than 0.05, it is not true. For the MRR the p-value of X5-Tool is higher (0.1078). This means that the Tool does not affect the MRR.
Response to comment 8: The statement regarding the discussion has been modified.
Comment 9: Section 5. The first sentence has no sense (line 252 – “The discharge energy generates between the workpiece and the toll.”). For sake of clarity, I suggest adding to Fig. 5 evidence of cracks, lumps and other details reported in the text.
Response to comment 9: The SEM images have been annotated.
Comment 10: What can you conclude from the morphological analysis? It is not correlated to different levels of the input, but it is in reported just as a function of the tool. This is because the aspect of the surface is the same varying the other parameters and the only effect that you can see is related to the tool? Did you detect recast particles? Different tool affect the presence of these particles?
Response to comment 10: Authors are grateful for this suggestion. It has been concluded from the morphological study that the copper electrode perform better than other two electrode material. The deposited layer are presented in the SEM micrograph. But the actual measurement of recast layer thickness was not made. In the future, we will definitely consider this aspect. However, it has been found that the tool material effect the presence of deposited layer, which is depicted in SEM images (section 5).
Round 2
Reviewer 2 Report
The authors respon satisfactorily to the reviewer's comments. However, the plagiarism software identifies 16% of similirities with others document. In my opinion, authors must modify the manuscript for avoiding this similarities, otherwise, the paper cannot be accepted for publication.
none
Author Response
Reviewer-2
Response to comment: Authors are thankful for this suggestion, as during the revision process, we took the help from different published literature. Now, similarity index is reduced in the manuscript up to 14%.
Round 3
Reviewer 2 Report
The manuscript has been revised in a proper way answering to reviewer's comments
none